# Ex vivo cortical circuits learn to predict and spontaneously replay temporal patterns

**Benjamin Liu & Dean V. Buonomano** [ID] ✉

It has been proposed that prediction and timing are computational primitives of neocortical microcircuits, specifically, that neural mechanisms are in place to allow neocortical circuits to autonomously learn the temporal structure of external stimuli and generate internal predictions. To test this hypothesis, we trained cortical organotypic slices on two temporal patterns using dual-optical stimulation. After 24-h of training, whole-cell recordings revealed network dynamics consistent with training-specific timed prediction. Unexpectedly, there was replay of the learned temporal structure during spontaneous activity. Furthermore, some neurons exhibited timed prediction errors as revealed by larger responses when the expected stimulus was omitted compared to when it was present. Mechanistically our results indicate that learning relied in part on asymmetric connectivity between distinct neuronal ensembles with temporally-ordered activation. These findings further suggest that local cortical microcircuits are intrinsically capable of learning temporal information and generating predictions, and that the learning rules underlying temporal learning and spontaneous replay can be intrinsic to local cortical microcircuits and not necessarily dependent on top-down interactions.

The ability to predict and prepare for external events is among the most important computations the brain performs. Timing is a critical component of prediction because it is often necessary to anticipate when future events will occur. Given the critical role of prediction and timing in perception, behavior, and cognition, it has been proposed that they are computational primitives of neocortical microcircuits[1–4]. Specifically, neural mechanisms are in place to allow local neocortical microcircuits to autonomously learn the temporal structure of external stimuli and generate internal predictions of when subsequent stimuli will arrive. Testing this hypothesis in vivo is challenging because local cortical microcircuits are difficult to study in the absence of upstream and downstream influences. The view that cortical microcircuits are, in effect, "designed" to learn and perform certain types of computations leads to the hypothesis that some simple computations can be observed and studied in reduced preparations. Indeed, prior studies have shown that in vitro (here, defined as dissociated cultures) and ex vivo (acute slices, cortical organotypic cultures, or organoids) circuits have the ability to perform simple forms of pattern recognition

and learning[5–11]. These reduced model systems provide crucial insights into the extent to which local cortical circuits, independent of broader brain systems, can perform and learn simple computations.

Here, we used cortical organotypic cultures as an ex vivo approach to determine whether neocortical circuits can autonomously learn to predict the temporal structure of chronically presented stimuli and study the underlying mechanisms. Cortical organotypic cultures are well-suited to bridge conventional in vitro and in vivo approaches as they preserve much of the local in vivo neocortical microarchitecture[12–18]. Indeed, spontaneous activity in neocortical organotypic slices converges to similar dynamic regimes observed in vivo. Specifically, they exhibit Up-state/Down-state transitions that reflect the well-balanced excitatory-inhibitory regimes critical to normal neocortical function[17,19–21]. It is well established that synaptogenesis occurs in organotypic slices, which is thought to be driven by homeostatic learning rules aimed at bringing network dynamics back to homeostatic setpoints[14,15,18,21]. There is no evidence of abnormal connectivity between cell types or of autapses (as observed in

Department of Neurobiology, Deparment of Psychology, and Psychology, Integrative Center for Learning and Memory, University of California, Los Angeles, Los Angeles, CA, USA. ✉e-mail: dbuono@ucla.edu

dissociated cultures)—indeed, recent studies of microcircuit connectivity in human neocortex have revealed similar connection patterns in organotypic and acute cultures of human neocortex. In addition, the synaptic learning rules observed in acute slices and in vivo are present in organotypic slices, indeed a number of early studies of synaptic plasticity were performed in organotypic cultures[22–29]. Finally, and critical to our goals here, organotypic slices coupled with optogenetics provide a tractable way to fully control the "sensory" experience and to study forms of learning that may take hours or days to develop.

Prediction and timing are active computations in that they rely on internally generated neural dynamics supported by recurrent connectivity[1–4]. Therefore, it is critical that any ex vivo preparation used to study prediction and timing exhibit self-propagating neural dynamic regimes. Importantly, organotypic cultures exhibit Up-states and emergent neural dynamics consistent with the balanced regimes of inhibition-stabilized networks[17,19,30–32]. Inhibition-stabilized networks are characterized by self-sustained neural dynamics driven by recurrent excitation and held in check by inhibition[20,33–35]. Thus, organotypic cultures provide a unique opportunity to study dynamic regimes associated with prediction, timing, and replay—which have typically only been studied in vivo. Replay is generally defined as the spontaneous reactivation of activity patterns during resting or sleep states that mirror the spatiotemporal structure of activity that occurred during prior learning or behavior[36–43]. These "offline" spontaneous reactivations of patterned activity observed during the recent waking experience is in itself a form of network-level learning and may serve a purpose in the consolidation of memories or information by further engaging neural plasticity and synaptic restructuring mechanisms.

To study the learning of network-level computations, specifically prediction, and timing, sparse subpopulations of cortical pyramidal neurons were transduced with either Channelrhodopsin2 or ChrimsonR. Training consisted of the presentation of trains of red and blue light pulses separated by either a short (10 ms) or long (370 ms) interval for 24 h. Following training, we observed robust differential dynamics evoked by red light alone that was consistent with prediction

and learned timing of the blue light pulses. Unexpectedly, we also observed spontaneous replay of the learned temporal patterns. Such structured spontaneous activity[44–46] and replay[38,40,47–49] parallels in vivo studies. Overall, our results demonstrate that neocortical circuits are autonomously able to learn to generate timed prediction errors and replay. Consistent with the hypothesis that prediction, timing, and replay are computational primitives of neocortical microcircuits.

## Results

### Dual-optical approach to study network-level learning in ex vivo cortical circuits

We first established a dual-optical stimulation approach that leveraged sparse expression of Channelrhodopsin2 (ChR2) and ChrimsonR (Chrim) in cortical pyramidal neurons (Fig. 1). To achieve sparse and differential expression of two opsins, we expressed ChR2 and Chrim under the Cre and FLP promoters, respectively (see Methods). Approximately 10–15% of total neurons expressed ChR2 or Chrim with no detectable overlap (Fig. 1D). Using whole-cell patch-clamp recordings, we confirmed that 5 ms pulses of blue light reliably induced single action potentials in ChR2-expressing (ChR2+) neurons. Similarly, 5 ms pulses of red light reliably induced single action potentials in Chrim-expressing (Chrim+) neurons. The red light did not produce detectable depolarization of ChR2+ neurons, but as expected, blue light could produce mild subthreshold depolarization of Chrim+ neurons[50,51] in the presence of glutamate receptor antagonists (Fig. 1B). As described below, our experimental design relies on the ability to differentially stimulate two subpopulations of neurons, thus any potential subthreshold crosstalk or low level of coexpression of opsins does not significantly influence our experimental protocol, which was designed with these potential constraints in mind.

Training protocols mirrored behavioral delay conditioning paradigms (Fig. 1A). A long train of red light pulses (440 ms, 25 Hz) represented the "conditioned stimulus" (CS), and a shorter but higher frequency train of blue light pulses (80 ms, 50 Hz) represented the "unconditioned stimulus" (US). Mirroring the CS-US interstimulus intervals (ISIs) of behavioral studies, we used two red-blue light (CS-

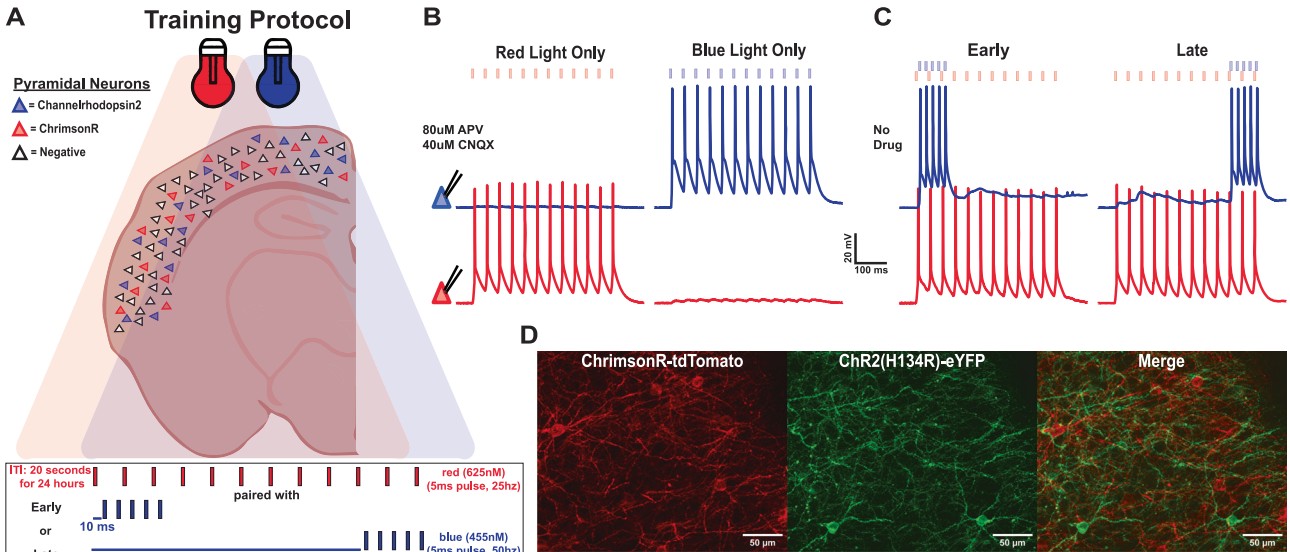

**Fig. 1 | A sparse dual-opsin approach for interval learning in ex vivo cortical circuits. A** Schematic of cortical pyramidal neurons in an organotypic slice culture sparsely transduced with either ChR2 or Chrim (top) and Early vs Late chronic optogenetic training paradigm (bottom). **B** Sample simultaneous whole-cell patch clamp recordings from two cortical pyramidal neurons expressing ChR2 (blue) or Chrim (red) during the presentation of red or blue light (5 ms, 12 pulses, 25 Hz) in the presence of synaptic blockers [80 μM APV, 40 μM CNQX]. **C** Sample

simultaneous recording from ChR2+ (blue) and Chrim+ (red) neurons during the presentation of the Early (left) and Late (right) training paradigms. **D** Image showing non-overlapping expression of ChR2 and Chrim in Layer 2/3 pyramidal neurons in the auditory cortex. Seven independent groups or organotypic slices were prepared at different time periods over the course of a year and trained in identical conditions, yielding consistent results; 3–4 mice per group with 6–8 cortical slices per mouse.

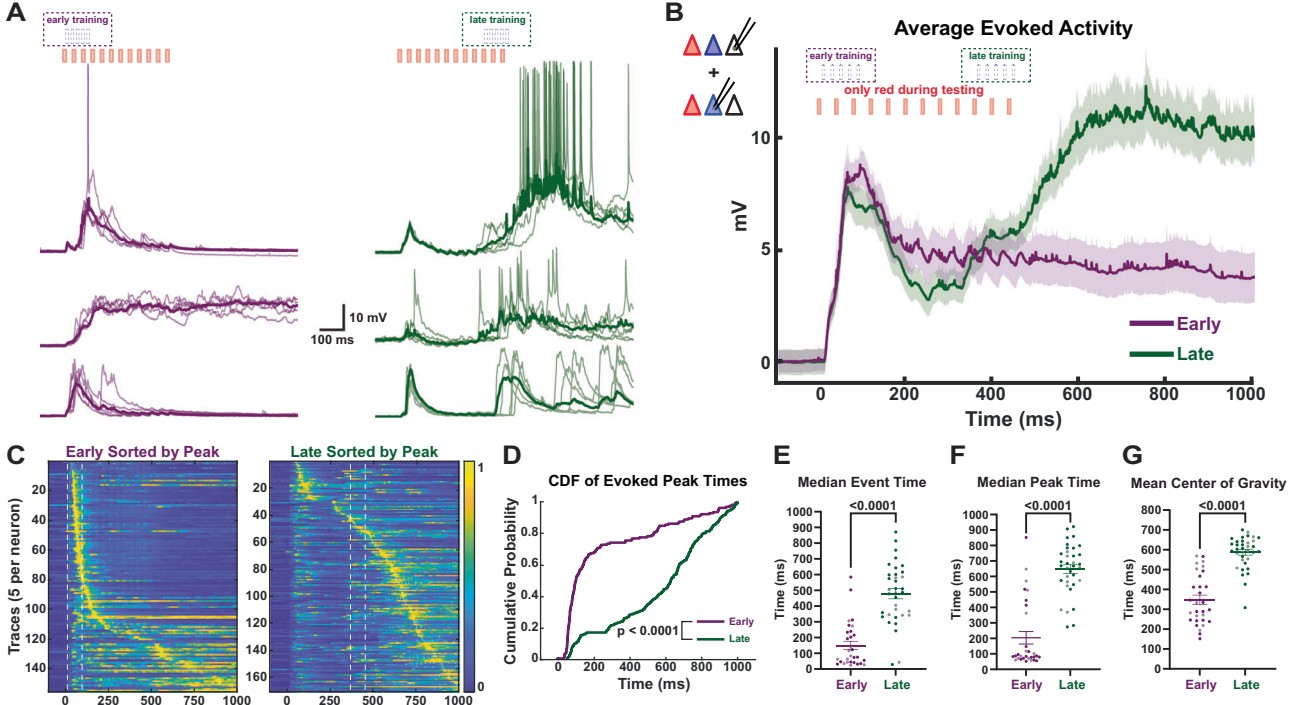

**Fig. 2 | The temporal profile of network dynamics is dependent on training interval. A** Traces of evoked polysynaptic activity from three sample neurons (5 traces per cell and mean (bold)) in response to red-alone in slices trained on the Early (purple) vs Late (green) paradigm. All samples were from Opsin⁻ neurons. **B** Comparison of the temporal profile of the mean ± SEM (shading) of evoked activity in pyramidal neurons from Early (Opsin⁻: 19, ChR2⁺: 11) vs Late (Opsin⁻: 24, ChR2⁺: 11) trained slices. **C** Normalized voltagegram of red light-evoked responses from ChR2⁺ and Opsin⁻ pyramidal neurons sorted by peak time in Early (left) vs Late (right) trained slices. Dashed white lines indicate when blue light stimulation occurred during training. **D** Cumulative distribution of evoked peak times was significantly different in recorded neurons from Early- vs Late-trained slices; $p < 10^{-8}$, Kolmogorov-Smirnov test. **E** Average median event time of evoked network activity was significantly lower in neurons from Early- vs Late-trained slices.

Opsin⁻ neurons are shown in dark purple or dark green, while ChR2⁺ neurons are shown in light purple or light green. Data are presented as median event times ± SEM: $150 \pm 26$ ms and $479 \pm 32$ ms for Early and Late, respectively; $U = 97$, $n_{\text{Early}} = 30$, $n_{\text{Late}} = 35$, $p < 10^{-9}$, two-sided Mann–Whitney test. **F** Average median peak time of evoked network activity was significantly lower in neurons from Early- vs Late-trained slices. Data are presented as median peak times ± SEM: $203 \pm 40$ ms and $648 \pm 27$ ms for Early and Late, respectively; $U = 77$, $n_{\text{Early}} = 30$, $n_{\text{Late}} = 35$, $p < 10^{-10}$, two-sided Mann–Whitney test. **G** Mean center of gravity of evoked network activity was significantly lower in neurons from Early- vs Late-trained slices. Data are presented as mean center of gravity ± SEM: $348 \pm 23$ ms and $586 \pm 14$ ms for Early and Late, respectively; $U = 61$, $n_{\text{Early}} = 30$, $n_{\text{Late}} = 35$, $p < 10^{-10}$, two-sided Mann–Whitney test. Source data for Figs. 2–6 are provided as a Source data file.

US) ISIs. In the Early condition, both stimuli were activated with similar onset times (10 ms ISI), while in the Late condition, the ISI was 370 ms, with similar offset times (Fig. 1C). Our primary goal was to test the hypothesis that following 24 h of chronic training, isolated neural circuits have the ability to learn to "predict" the presentation of the blue light at the appropriate time.

## Cortical circuits learn the temporal structure of experienced patterns

To test whether the cortical circuits were able to successfully learn the trained intervals, we initially recorded the responses of opsin-negative (Opsin⁻) and ChR2⁺ pyramidal neurons to the presentation of the red light pulses alone (red-alone). Following 24 h of training, recordings revealed differentially timed network dynamics in response to the red light pulses, which closely aligned with the corresponding training interval to which the slice was exposed. Specifically, in the Late, but not in the Early condition, presentation of red-alone generally elicited a marked late peak in network activity, suggesting the prediction of an expected (but absent) arrival of blue light stimulation (Fig. 2A). Qualitatively, both the averaged (Fig. 2B) and individual traces (Fig. 2C) revealed a large difference in temporal structure between the Early- and Late-training groups. These differences were confirmed by quantification of the temporal distribution of the polysynaptic events within each trace (see Methods) between Early- and Late-trained neurons (Kolmogorov-Smirnov test, $p < 10^{-8}$;

Fig. 2D). The median times of all the detected polysynaptic events from a neuron were also significantly different ($150 \pm 26$ ms and $479 \pm 32$ ms for Early and Late, respectively; $U = 97$, $n_{\text{Early}} = 30$, $n_{\text{Late}} = 35$, $p < 10^{-9}$, Mann–Whitney test; Fig. 2E). Similarly, the time of the peak postsynaptic potential was significantly different between the Early and Late groups, with median latencies of $203 \pm 40$ ms and $648 \pm 27$ ms respectively ($U = 77$, $n_{\text{Early}} = 30$, $n_{\text{Late}} = 35$, $p < 10^{-10}$, Mann–Whitney test; Fig. 2F). Lastly, the time of the center of gravity of the mean response of each cell was also significantly different between neurons in Early- vs Late-trained slices ($U = 61$, $n_{\text{Early}} = 30$, $n_{\text{Late}} = 35$, $p < 10^{-10}$, Mann–Whitney test; Fig. 2G). Together, these results demonstrate that the temporal profile of evoked neural activity was differentially shaped in a training-dependent manner. This finding suggests that isolated cortical circuits are intrinsically capable of learning and predicting the temporal structure of experienced stimuli.

Notably, while the temporal profile of evoked activity was distinct between neurons in the Early- vs Late-training conditions, no significant differences were observed between simultaneously recorded ChR2⁺ and Opsin⁻ neurons within the same training condition (Supplementary Fig. S1). However, consistent with previous results[52], the amplitude of the evoked activity was larger in the Opsin⁻ compared to the ChR2⁺ neurons. Consistent with the notion that training-dependent learning is a network-wide phenomenon (see below).

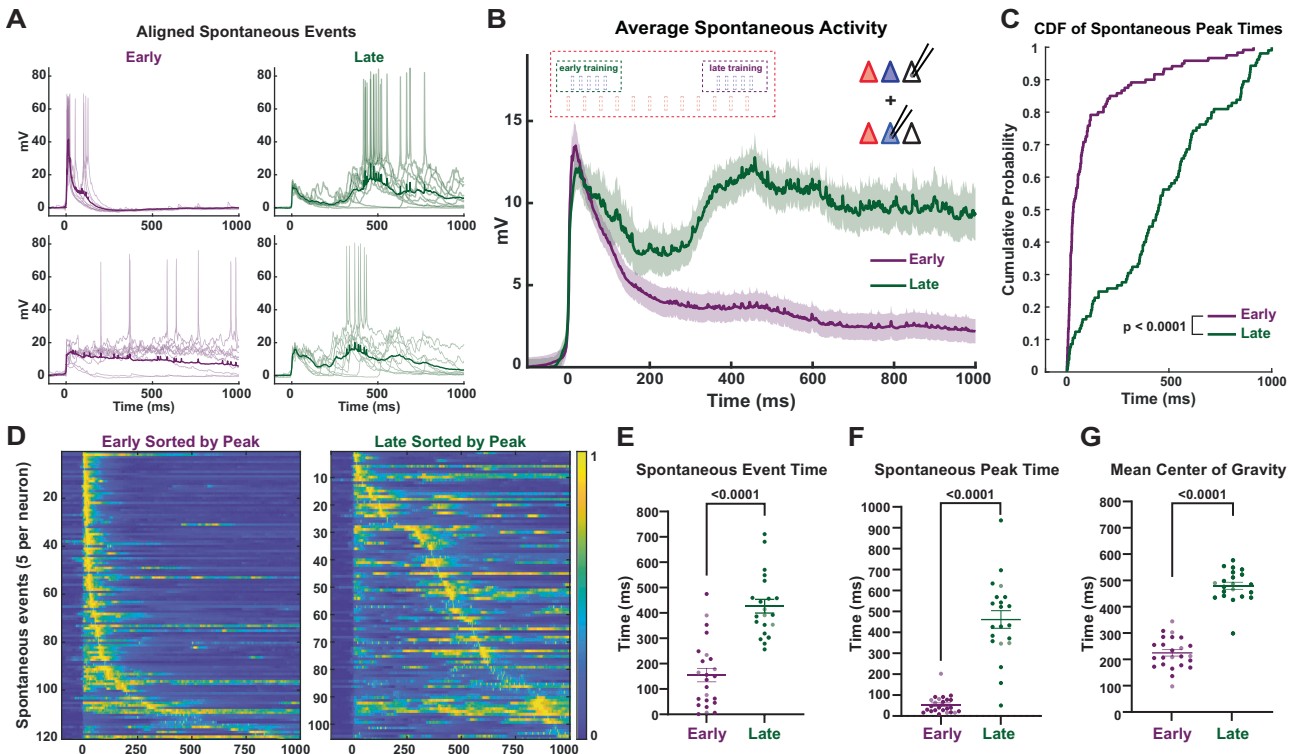

**Fig. 3 | Spontaneous network activity mirrors the learned interval-specific network dynamics. A** Sample traces of aligned spontaneous network activity (10 spontaneous events per cell and mean (bold)) from Opsin⁻ pyramidal neurons in Early (purple) and Late (green) trained slices. **B** Comparison of the mean ± SEM (shading) of spontaneous activity in pyramidal neurons from Early (Opsin⁻: 18, ChR2⁺: 6) vs Late (Opsin⁻: 18, ChR2⁺: 3) trained slices. **C** Cumulative distribution of the spontaneous event peak times in ChR2⁺ and Opsin⁻ pyramidal neurons from Early- vs Late-trained slices; $p < 10^{-6}$, Kolmogorov-Smirnov test. **D** Normalized voltagegram of spontaneous events from recorded pyramidal neurons sorted by peak time in Early (left) vs Late (right) trained slices. **E** Average median event times of spontaneous events was significantly lower in neurons from Early- vs Late-trained

slices. Opsin⁻ neurons are shown in dark purple or dark green, while ChR2⁺ neurons are shown in light purple or light green. Data are presented as median event times ± SEM: 155 ± 26 ms and 427 ± 27 ms for Early and Late, respectively; $U = 36$, $n_{Early} = 24$, $n_{Late} = 21$, $p < 10^{-7}$, two-sided Mann−Whitney test. **F** Average median peak times of spontaneous events was significantly lower in neurons from Early- vs Late-trained slices. Data are presented as median peak times ± SEM: 54 ± 9 ms and 462 ± 42 ms for Early and Late, respectively; $U = 11$, $n_{Early} = 24$, $n_{Late} = 21$, $p = 10^{-10}$, two-sided Mann−Whitney test. **G** Mean center of gravity of spontaneous events was significantly lower in neurons from Early- vs Late-trained slices. Data are presented as mean center of gravity ± SEM: 230 ± 12 ms and 483 ± 13 ms for Early and Late, respectively; $U = 3$, $n_{Early} = 24$, $n_{Late} = 21$, $p < 10^{-10}$, two-sided Mann−Whitney test.

## Cortical circuits spontaneously replay learned dynamics

As previously described, organotypic cultures exhibit bouts of spontaneous activity, which can be homeostatically regulated[13,17,52]. Unexpectedly, the temporal profile of these spontaneous bouts of activity were different between training groups. Moreover, the temporal structure of the spontaneous activity closely mirrored the learned, training-dependent, evoked network dynamics (Fig. 3A). Group averaged data revealed a robust training-dependent difference in the temporal structure of spontaneous events (Fig. 3B). To quantify these differences, we isolated bouts of spontaneous activity and analyzed the temporal structure time-locked to the onset of each bout (see Methods). This was confirmed by multiple measures: differences in the distribution of peak time of the spontaneous bouts of activity (Kolmogorov-Smirnov test, $p < 10^{-6}$; Fig. 3C, D), the median time of the detected polysynaptic events ($U = 36$, $n_{Early} = 24$, $n_{Late} = 21$, $p < 10^{-7}$, Mann−Whitney test; Fig. 3E), the time of the spontaneous peak of the postsynaptic potential ($U = 11$, $n_{Early} = 24$, $n_{Late} = 21$, $p = 10^{-10}$, Mann−Whitney test; Fig. 3F), and the mean center of gravity of spontaneous events ($U = 3$, $n_{Early} = 24$, $n_{Late} = 21$, $p < 10^{-10}$, Mann−Whitney test; Fig. 3G).

In order to establish, on a cell-by-cell case, that the training-specific temporal profile of evoked and spontaneous activity were correlated, we directly compared the evoked and spontaneous activity dynamics (Fig. 4). As shown in the comparison of two different sample neurons from each training group, the average evoked

and spontaneous activity was qualitatively similar (Fig. 4A), and consistent with the notion that the neural trajectories evoked by red light were being spontaneously replayed "offline". This was confirmed by the similarity between the distributions of spontaneous and evoked peak times across cells within one training condition, and the significant difference between the timing of evoked Early vs Late events (Kolmogorov-Smirnov test, $p < 10^{-8}$) and spontaneous Early vs Late events (Kolmogorov-Smirnov test, $p = 10^{-7}$; Fig. 4B). To further quantify the similarity between spontaneous and evoked activity within training conditions, we computed the correlation coefficients between mean evoked and mean spontaneous activity across all recorded neurons within a training group, excluding within cell comparisons (Fig. 4C, top row). Furthermore, we also computed the correlation coefficients between the mean evoked and mean spontaneous activity across all recorded neurons between training groups (Fig. 4C, bottom row). Interestingly, the correlation coefficients of mean evoked and mean spontaneous activity was much higher across neurons from different slices within a training condition compared to across training conditions ($U = 19342$, Early Evoked vs Early Spont = 552, Early Evoked vs Late Spont = 504, $p < 10^{-10}$, Mann−Whitney test) ($U = 45680$, Late Evoked vs Late Spont = 504, Late Evoked vs Early Spont = 420, $p < 10^{-10}$, Mann−Whitney test; Fig. 4D). Thus, even across cortical circuits from different slices, the temporal structure of the evoked and spontaneous activity appeared to be similarly shaped by the training paradigm to which the circuits were

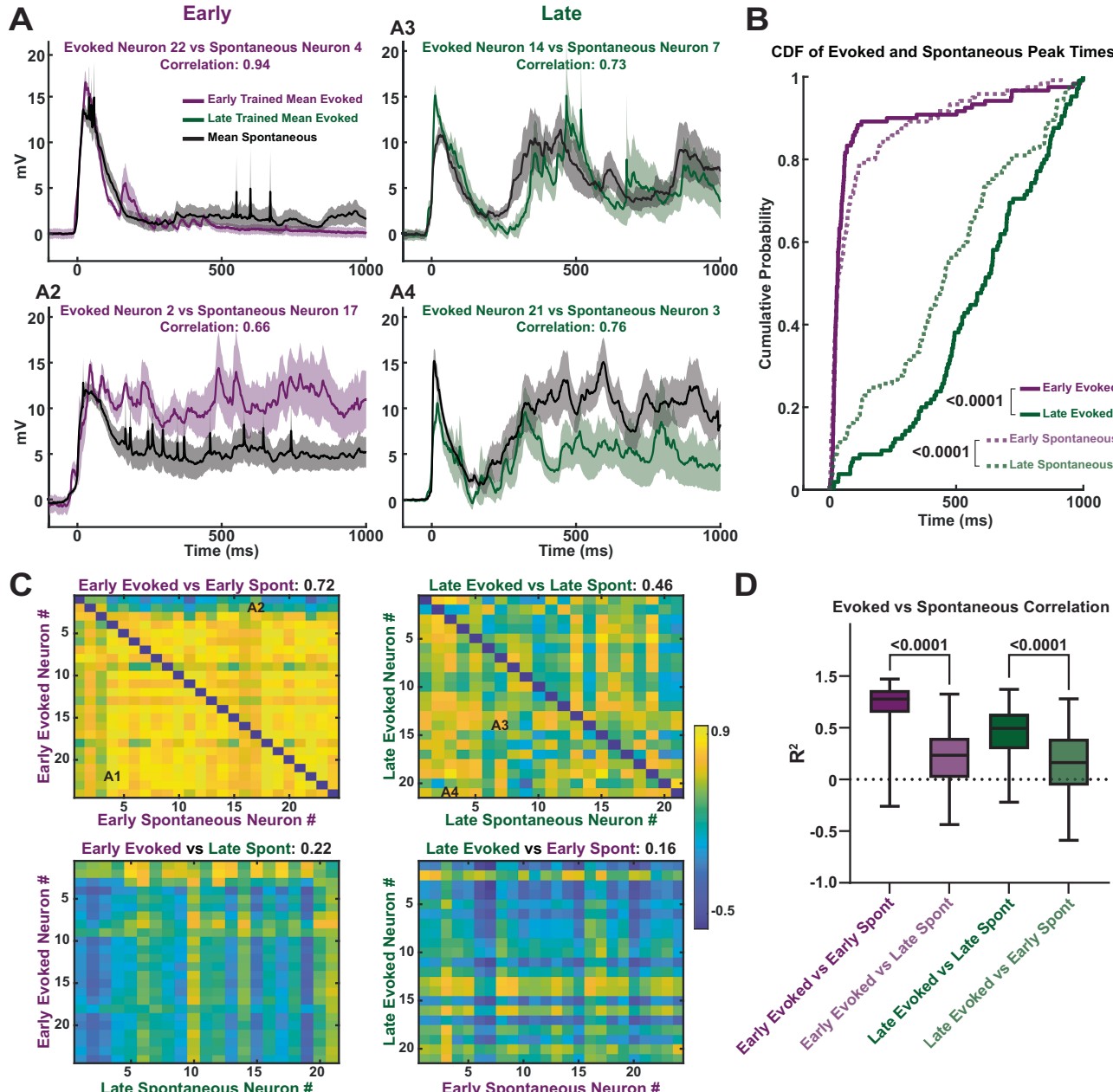

**Fig. 4 | Training-specific replay demonstrated by the higher correlation between evoked and spontaneous network activity across slices. A** Sample mean ± SEM (shading) of evoked (Early: purple, Late: green) vs spontaneous (black) network activity from pyramidal neurons in different slices within the same training group. **B** Comparison of the cumulative distribution of evoked and spontaneous activity peak times across slices between training groups. Early evoked vs Late evoked; $p < 10^{-8}$, Kolmogorov-Smirnov test. Early spontaneous vs Late spontaneous; $p = 10^{-7}$, Kolmogorov-Smirnov test. **C** Correlation matrix of mean evoked vs mean spontaneous activity across neurons within the same training group (top) and across training groups (bottom). **D** Box plot of correlation values for mean evoked vs mean spontaneous activity across neurons both within and across training groups (median, max, min, 1st, and 3rd quartile displayed). (Early Evoked vs Early Spontaneous) vs (Early Evoked vs Late Spontaneous); $n = 552$, $n = 504$; $p < 10^{-10}$, two-sided Mann–Whitney test. (Late Evoked vs Late spontaneous) vs (Late Evoked vs Early Spontaneous); $n = 420$; $n = 504$; $p < 10^{-10}$, two-sided Mann–Whitney test.

exposed, suggestive of specific neural trajectories being "burned-in" to the circuits as a result of training.

## Distinct neuronal ensembles with temporally-ordered activation

The above results reveal that local cortical microcircuits learn to reproduce the temporal structure of the patterned stimulation they experience during training by sculpting the temporal profile of network dynamics. To elucidate the network-level mechanisms that underlie the learned temporal dynamics, we reasoned that potential connectivity differences between subpopulations of neurons may be revealed by the cross-correlation structure between pairs of neurons during spontaneous activity. During Late-training in the incubator, Chrim+ neurons were consistently activated hundreds of milliseconds before ChR2+ neurons. We thus asked if this was also true during spontaneous replay by performing simultaneous whole-cell recordings of Chrim+ and ChR2+ expressing neurons (Fig. 5). As expected, in Untrained circuits, spontaneous activity in Chrim+ and ChR2+ neurons was highly correlated, but in Late-trained circuits this correlation was significantly weaker, suggesting that the Chrim+ and ChR2+

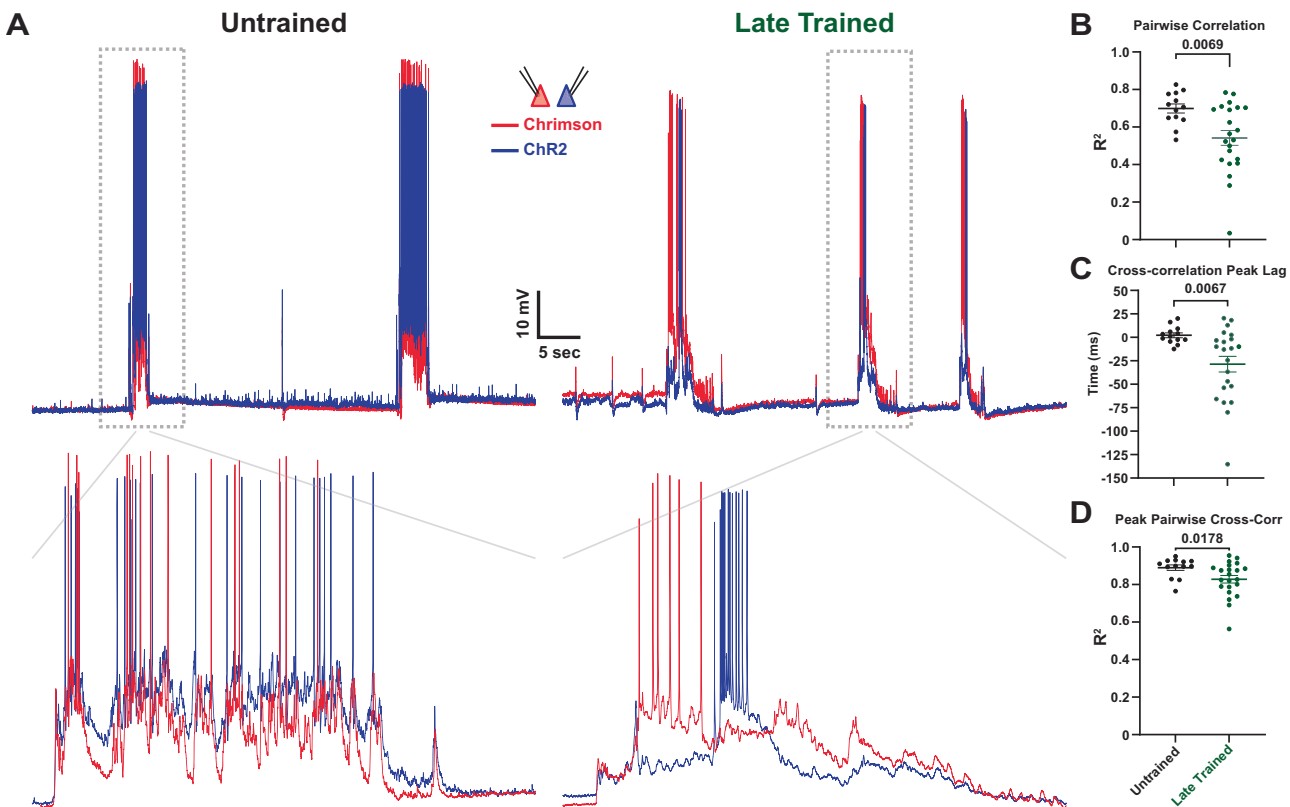

**Fig. 5 | Training-dependent emergence of distinct neuronal ensembles as indicated by the decorrelation and temporal lag between Chrim⁺ and ChR2⁺ neurons. A** Sample 60 s trace of spontaneous network activity from simultaneously recorded ChR2⁺ (blue) and Chrim⁺ (red) neurons (top). The dashed box is magnified in the traces below (2.5 s window, bottom). **B** Significant decrease in the average ± SEM pairwise voltage correlation of spontaneous events in simultaneously recorded ChR2⁺ and Chrim⁺ neurons in Late-trained compared to Untrained slices. Mean $r^2$ = Untrained: 0.71, Late: 0.55; $U = 13$, $n_{Untrained} = 13$, $n_{Late} = 22$, $p = 0.007$, two-sided Mann–Whitney test. **C** Average ± SEM cross-correlation of spontaneous events in simultaneously recorded ChR2⁺ and Chrim⁺ neurons revealed a consistent temporal lag in ChR2⁺ compared to Chrim⁺ neurons. $U = 65$, $n_{Untrained} = 13$, $n_{Late} = 22$, $p = 0.007$, two-sided Mann–Whitney test. Sample spontaneous event with temporal lag in simultaneously recorded ChR2⁺ and Chrim⁺ neurons from a Late-trained slice (Panel **A**, bottom right). **D** Average ± SEM peak pairwise cross-correlation of spontaneous events is less significantly different than the average pairwise correlation of spontaneous events in simultaneously recorded ChR2⁺ and Chrim⁺ neurons. Mean $r^2$ = Untrained: 0.90, Late: 0.84; $U = 74$, $n_{Untrained} = 13$, $n_{Late} = 22$, $p = 0.018$, two-sided Mann–Whitney test.

populations may have formed into distinct neuronal ensembles as a result of experience ($r^2$ = Untrained: 0.71, Late: 0.55; $U = 13$, $n_{Untrained} = 13$, $n_{Late} = 22$, $p = 0.007$, Mann–Whitney test; Fig. 5A, B). Cross-correlation analyses revealed a temporal lag in the peak cross-correlation of spontaneous activity dynamics between Chrim⁺ and ChR2⁺ neurons ($31 \pm 11$ ms; Fig. 5C), with the Chrim⁺ neurons leading the ChR2⁺ (Fig. 5A, right). In contrast, in the Untrained group, the peak cross-correlation lag between ChR2⁺ and Chrim⁺ neurons was approximately zero ($U = 65$, $n_{Untrained} = 13$, $n_{Late} = 22$, $p = 0.007$, Mann–Whitney test; Fig. 5C). This temporally ordered activation of Chrim⁺ → ChR2⁺ neurons suggested training-dependent changes in the synaptic connectivity of Chrim⁺ and ChR2⁺ neurons. This was further supported by the smaller difference in peak pairwise cross-correlation of Chrim⁺ x ChR2⁺ spontaneous events ($U = 74$, $n_{Untrained} = 13$, $n_{Late} = 22$, $p = 0.018$, Mann–Whitney test; Fig. 5D) compared to the pairwise correlation of Chrim⁺ x ChR2⁺ in Late-trained slices.

## Asymmetric connectivity between different ensembles of excitatory neurons

In order to understand the temporal asymmetry between the different subpopulations of excitatory neurons, we next analyzed the monosynaptic connectivity between different populations. Due to the relatively sparse transduction, Chrim⁺ and ChR2⁺ neurons were generally far apart (>100 μm), making it difficult to study synaptic connectivity, which drops off dramatically with distance[53]. However, since the temporal profile of evoked activity in the ChR2⁺ and Opsin⁻ neurons was similar (Supplementary Fig. S1), we assessed connectivity between pairs of neighboring (<50 μm) Chrim⁺ and Opsin⁻ neurons in Layer 2/3 (Fig. 6A, B). Following Late-training, we observed a significant bias in the direction of synaptic strength, with the connections from Chrim⁺→Opsin⁻ neurons being stronger than the connections from Opsin⁻→Chrim⁺ neurons (Fig. 6C). We did not detect a significant difference in the directionality of connection probability (Fig. 6D). This finding suggests that the ability to learn and predict temporal patterns may depend in part on the training-induced asymmetry in excitatory synaptic strength between distinct neuronal ensembles.

## Prediction or temporal prediction errors?

The above results establish that neocortical microcircuits are autonomously capable of learning to not only predict external events but predict when those events are expected to occur. These results are broadly consistent with computational and in vivo studies suggesting that some forms of prediction and timing are computational primitives. In the Late-training group, our results demonstrate that, on average, the learned late response starts to emerge at approximately the time of the expected blue light onset during training, but only peaks after the expected offset of the blue light (Fig. 2B). Interestingly, this internally generated late response could be interpreted as either prediction/anticipation of the expected arrival of blue light, or as a prediction error generated by the absence of blue light. The distinction between prediction and prediction errors can be understood by comparing the evoked response to the red light alone (CS-only) and

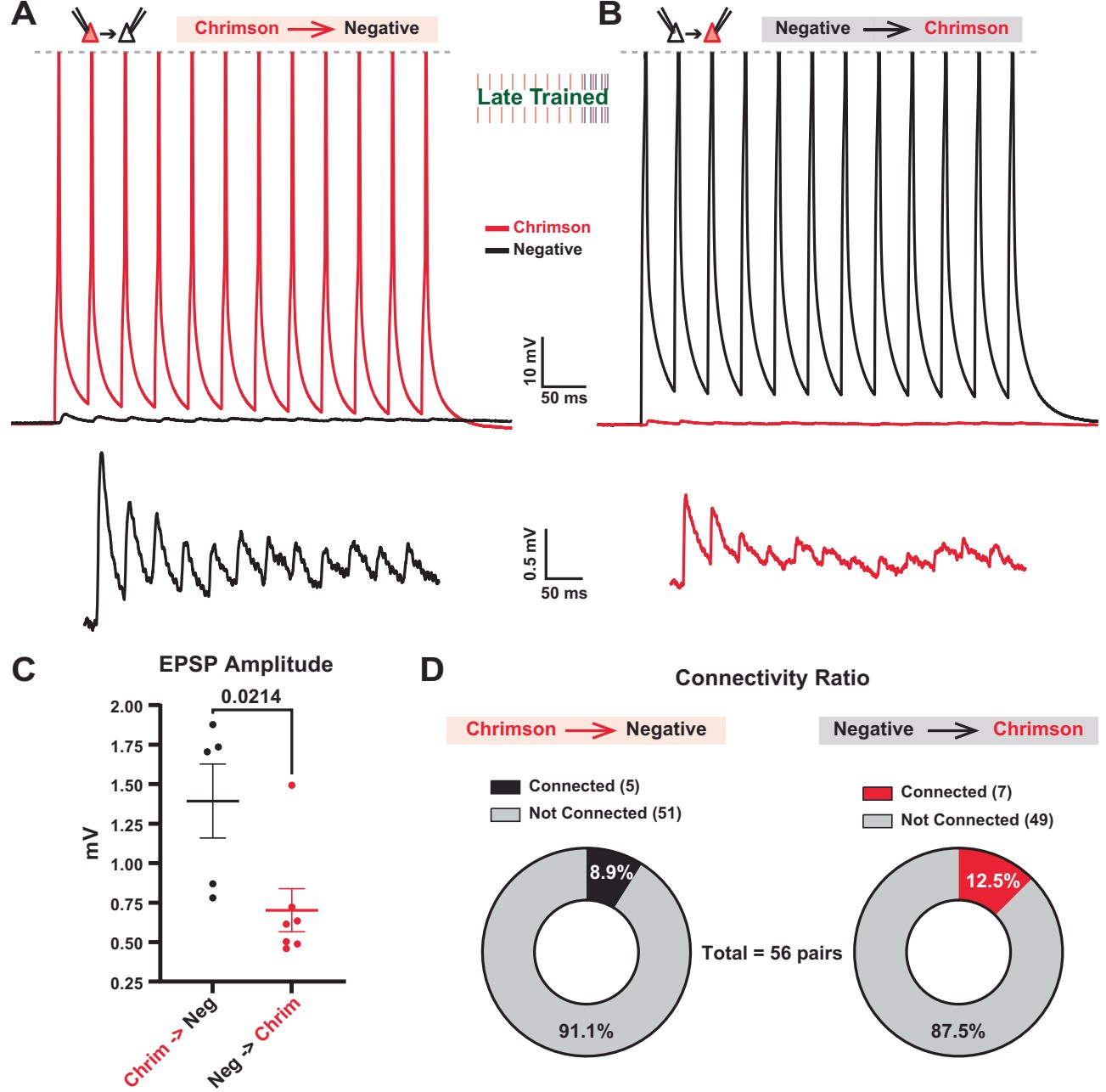

**Fig. 6 | Asymmetric connectivity between different subpopulations of excitatory neurons. A** Trial averaged traces from a paired recording between connected Chrim⁺ (red) →Opsin⁻ (black) pyramidal neurons (top) and an enlarged view of the EPSPs (bottom). **B** Trial averaged traces from a paired recording between connected Opsin⁻→Chrim⁺ pyramidal neurons (top) and an enlarged view of the EPSPs (bottom). **C** Mean ± SEM EPSP amplitudes of synaptic connections were significantly stronger in the direction of Chrim⁺→Opsin⁻ connections compared to Opsin⁻→Chrim⁺ connections; $n = 5$, $n = 7$, $p = 0.021$, two-sided unpaired $t$ test. **D** Connection probability was not significantly different in the Chrim⁺→Opsin⁻ direction compared to the Opsin⁻→Chrim⁺ direction.

red + blue light (CS-US). Specifically, if the late responses represent prediction errors, the responses to red + blue light should actually be weaker than the responses to red-alone. We thus compared responses to red-alone and red + blue after 24-h of Late-training. Indeed, some neurons responded more to red-alone than red + blue stimulation, as would be expected from prediction error neurons (Fig. 7A, B)—in effect, these cells exhibited an internally generated late response that was inhibited by blue light. Other neurons exhibited approximately equal responses to both red-alone and red + blue light (Fig. 7C, D). Overall, approximately a fourth of the neurons exhibited responses consistent with prediction errors (Fig. 7E). There was no significant difference in the input resistance (182 ± 11 and 187 ± 14 MΩ),

membrane time constant (10.4 ± 0.67 and 10.3 ± 0.37 ms), or intrinsic excitability (number of spikes per current step ranging from 0.05-0.3 nA) between the neurons that exhibited timed prediction errors and those that did not. But future studies will have to examine whether the observed distinction between prediction error and non-prediction error neurons correspond to specific excitatory neuron subtypes.

## Discussion

The ability to predict when events in the external world will occur is a fundamental component of animal intelligence, as it provides a means to anticipate and prepare for external events before they happen, efficiently encode expected information, and rapidly attend to

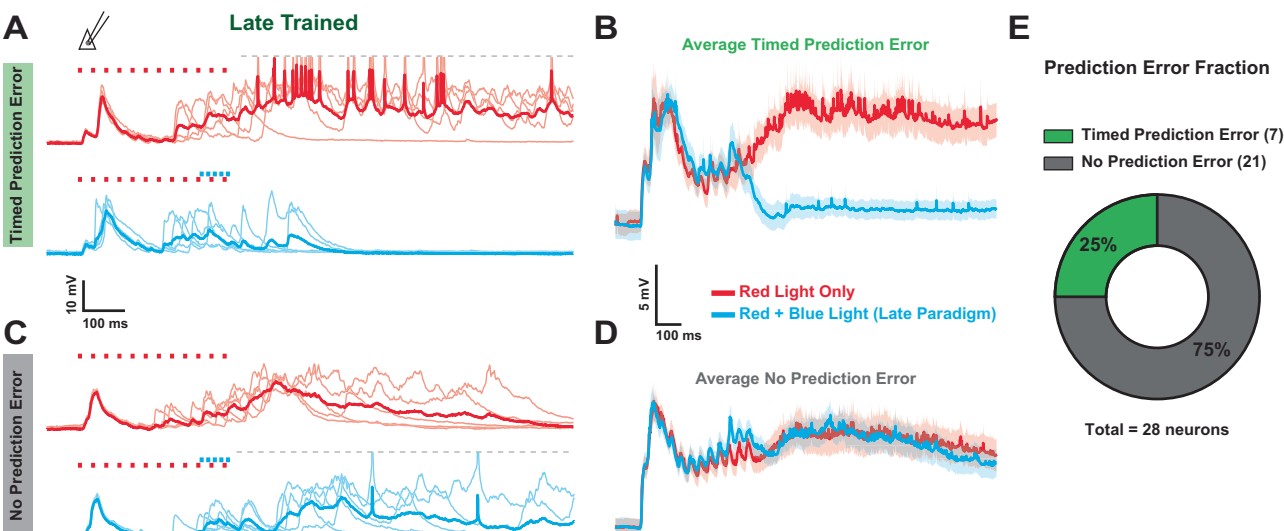

**Fig. 7 | Following Late-training a subpopulation of Opsin⁻ pyramidal neurons showed responses consistent with a timed prediction error. A** Five sample traces and mean (bold) from an Opsin⁻ pyramidal neuron with larger responses to red-alone compared to red + blue light (Late-training protocol). **B** Average responses ± SEM (shading) from Opsin⁻ pyramidal neurons expressing timed prediction error responses; *n* = 7. **C** Five sample traces and mean trace (bold) from an Opsin⁻ pyramidal neuron with similar responses to both red-alone and red + blue light. **D** Average responses ± SEM (shading) from Opsin⁻ pyramidal neurons that did not express timed prediction error responses; *n* = 21. **E** Ratio of pyramidal neurons with timed prediction error responses (25%) vs no prediction error responses (75%).

surprising events. Because of the importance of prediction and timing, a wide range of learning and neurocomputational theories, including classical and operant conditioning, reinforcement learning, predictive coding, and Bayesian inference, are based on the ability to learn to make predictions based on previous experiences[3,4,54–57]—some of these theories explicitly address the problem of time while in others the temporal component of prediction is largely absent. Given the importance of prediction and timing, it has been proposed that they both may be computational primitives of neocortical circuits[2–4,58]. Consistent with this view, it has been shown that visual cortex circuits can learn to predict the timing of an expected reward[59,60]. Similarly, in the barrel cortex, it has been shown that neurons develop internally generated responses that peak at approximately the onset time of an expected sensory event[61] and that some neurons exhibited larger responses if the expected sensory event arrived late—a finding that can be interpreted as a prediction error.

In the context of predictive coding theories, it is often assumed that predictive responses and prediction errors in sensory areas always require top-down signals[3,4,62]. However, a few studies have also observed simple forms of timed prediction in acute and organotypic slices[6,7,11]. Here, by using a dual-optical training approach, we were able to directly address whether individual neurons in "standalone" neocortical circuits were able to learn to generate predictive responses and prediction errors. Our approach also allowed for the explicit identification of subpopulations of neurons representing both sensory stimuli, and to perform paired intracellular recordings to analyze the connectivity patterns between these subpopulations.

The identification of single neurons that show prediction error responses provides direct evidence for predictive coding in a reduced and highly controlled cortical network. The large response observed to the omission of the expected blue light can also be characterized as an omission response. And the fact that these responses are delayed, peaking approximately 265 ± 39 ms after the expected onset of the blue light, further indicated that these responses rely on complex changes in the internal circuitry rather than simple blue-light inhibition.

Our results establish that neocortical circuits are autonomously able to learn the temporal structure of the stimuli to which they are exposed. Furthermore, we observed that during spontaneous activity, the temporal profile of network dynamics reproduced the temporal structure of the training protocol and post-training evoked activity. This resemblance in the temporal structure of evoked and spontaneous activity post-training implies that a large degree of overlap may exist between the participating neurons in either form of activity, as well as a similar spatiotemporal order of activation. This finding provides evidence for a form of learned replay in ex vivo cortical circuits.

### Neural mechanisms underlying prediction and timing
Previous studies have demonstrated that ex vivo cortical circuits can exhibit reproducible network-wide patterns of internally generated activity. Such self-generated patterns can take numerous forms, including Up-states and time-varying neural trajectories[19,44,63,64]. Here, we show that these internally generated neural trajectories are shaped by experience in a manner that encodes predictive timing. However, the neural and synaptic mechanisms underlying these neural trajectories remain unknown. A necessary step towards understanding the neural mechanisms underlying the learned network dynamics observed here is to characterize the differential dynamics of distinct subpopulations of neurons. As a step in this direction, paired whole-cell recordings from Chrim⁺ and ChR2⁺ positive neurons from Late-trained slices revealed a clear temporal order in their firing patterns during spontaneous activity (Fig. 5). Specifically, the peak cross-correlation lag of both the subthreshold voltage and spikes revealed that Chrim⁺ neurons were active before ChR2⁺ neurons. This observation is consistent with the notion that training resulted in the emergence of "red" and "blue" neuronal ensembles and that the "red" ensemble drove activity in the "blue" ensemble.

While the temporal profile of activity in the Chrim⁺ and ChR2⁺ neurons were distinct, we did not observe any significant differences in the temporal profile between ChR2⁺ and Opsin⁻ neurons (Supplementary Fig. S1) during either evoked or spontaneous activity. Suggesting that Chrim⁺ neurons may also drive Opsin⁻ neurons. To address this question, we performed connectivity analyses between Chrim⁺ and Opsin⁻ negative neurons—as mentioned, it was not possible to obtain connectivity data between Chrim⁺↔ChR2⁺ neurons because of the distance between sparsely transduced neurons. Based on the average synaptic strength of the monosynaptically connected pairs, we observed an asymmetry between the Chrim⁺ and Opsin⁻ pyramidal

subpopulations: synaptic connections were stronger in the Chrim$^+$→Opsin$^-$ direction compared to the Opsin$^-$→Chrim$^+$ direction (Fig. 6). These results provide direct evidence for the experience-dependent formation of asymmetric connectivity patterns in the neocortex.

As mentioned above, most theories of predictive coding propose that prediction error signals rely on feedback from higher-order cortical areas[65–68]. The current results support the possibility that each local neocortical microcircuit is autonomously able to generate prediction errors[69]—which, of course, does not imply that feedback is not also critical. And, of course, it remains to be determined if local neocortical circuits in vivo are autonomously capable of learning temporal predictions in the absence of interareal and top-down connections.

We propose that these changes in excitatory connections between different neuronal ensembles contribute to learning and timed prediction. However, based on previous experimental and computational work we also suggest that internally generated dynamics require orchestrated plasticity operating at both excitatory and inhibitory synapses[7,35]. Indeed, because learning requires the presence of internally generated neural dynamics that rely on positive excitation held in check by inhibition, we hypothesize that the learned dynamics operate in an inhibition-stabilized regime[20,34,70,71]. Thus, future studies should be aimed at studying the reciprocal connectivity between excitatory and inhibitory neurons, as well as training-specific and dynamic changes in the balance of excitation and inhibition.

## Methods

Animal ethics guidance and research protocol approval was provided by the UCLA Animal Research Committee.

### Organotypic cultures

Cortical organotypic slice cultures were prepared as described previously[7,13,52]. Slices were obtained from postnatal day 5–7 wild-type FVB mice of either sex. Organotypic cultures were prepared using the interface method[72]. Coronal slices (400 μm thickness) containing primary auditory and somatosensory cortex were sectioned using a vibratome (Leica VT1200) and bisected before being individually placed onto Millicell cell culture inserts (MilliporeSigma) in a 6-well plate with 1 mL of culture media per well. Culture media was changed at 1 and 24 h after initial plating and every 2–3 days thereafter. Cutting media consisted of MEM (Corning 15-010-CV) plus (final concentration in mM): MgCl$_2$, 3; glucose, 10; HEPES, 25; and Tris-base, 10. Culture media consisted of MEM (Corning 15-010-CV) plus (final concentration in mM): glutamine, 1; CaCl2, 2.6; MgSO4, 2.6; glucose, 30; HEPES, 30; ascorbic acid, 0.5; 20% horse serum, 10 units/L penicillin, and 10 μg/L streptomycin. Slices were incubated in 5% CO2 at 35 °C.

### Viral transduction

For the double sparse transduction, slices were transduced with a total of 4 viruses: AAV9-CamKII(0.4)-Cre (Addgene plasmid #105558), AAV9-EF1a-DIO-hChR2(H134R)-EYFP (Addgene plasmid #20298), AAV8-CAG-FLPX-ChrimsonR-tdTomato (Addgene plasmid #130909), and AAV9-CamKII(0.4)-FLPo (Vector biolabs). All 4 viruses had a starting titer of approximately [$1 \times 10^{13}$] and were combined into a viral cocktail before delivery to the slices. First, the two recombinase-expressing viruses CamKII-Cre and CamKII-FLP, were individually diluted with nuclease-free water to a concentration of approximately [$1 \times 10^{11}$]. The two diluted recombinase viruses were then combined in a 1:1 ratio by volume. The two undiluted opsin viruses, DIO-hChR2 and FLPx-ChrimsonR were also combined in a 1:1 ratio by volume. The combined diluted recombinase viruses were then further diluted by adding the combined opsin viruses in a 1:3 ratio by volume. The resulting final concentrations for all 4 viruses were approximately: [$1 \times 10^{10}$] AAV9-CamKII-Cre, [$1 \times 10^{10}$] AAV9-CamKII-FLPo, [$8 \times 10^{12}$] AAV9-DIO-hChR2, [$8 \times 10^{12}$] AAV8-FLPx-ChrimsonR. Each hemi-slice received a total of

0.8 μL of viral cocktail gently delivered via a sterilized pipette above the cortex. All viral transductions were performed at day-in-vitro (DIV) 7, and recordings were performed between DIV 22–30 to allow sufficient time for viral expression.

### Chronic optogenetic training

To minimize variability, experiments relied on "sister" slices, i.e., experimental batches were collected on the same day from the same litter of animals, maintained with the same culture medium/serum, placed in the same incubator, and virally transduced in the same session. For the interval-training experiments, both Early/Late-trained and Untrained control slices received equal amounts of virus and were simultaneously placed into the training incubator to ensure identical environments and experimental conditions. In addition, experiments were balanced by training and recording from an equal number of sister slices from each experimental condition per day. For chronic optical training, individual cell culture inserts containing one hemi-slice were placed in 6-well plates and quickly transferred along with their sister slices from the culture incubator to an identical "training incubator", where each individual slice is aligned with a dual-channel RGB LED (Vollong part #: VL-H01RGB00302).

Both Early- and Late-training protocols consisted of a 440 ms train of red light pulses (625 nm, 12 pulses, 5 ms each, 25 Hz, 0.2 mW/mm$^2$) paired with an 80 ms train of blue light pulses (455 nm, 5 pulses, 5 ms each, 50 Hz, 0.15 mW/mm$^2$) at two different temporal relationships. In the Early-training case, the onset of the train of red light pulses preceded the onset of the train of blue light pulses by 10 ms, while in the Late training case, the onset of red preceded the onset of blue by 370 ms. In both training cases, patterned optical stimulation was delivered every 20 seconds for approximately 24 h (±2 h). Following 24 h of training, slices were individually transferred from the training incubator to the whole-cell patch clamp rig for recordings between 1–6 h after the cessation of patterned stimulation.

### Electrophysiology

Cell culture inserts were transferred to the recording rig and perfused with oxygenated ACSF composed of (mM): 125 NaCl, 5.1 KCl, 2.6 MgSO$_4$, 26.2 NaHCO$_3$, 1 NaH$_2$PO$_4$, 25 glucose, 2.6 CaCl$_2$ (ACSF was formulated to match the standard culture media). Temperature was maintained at 32–33 °C and perfused at 5 mL/min. The Whole-cell solution was composed of (mM): 100 K-gluconate, 20 KCl, 4 ATP-Mg, 10 phosphocreatine, 0.3 GTP, 10 HEPES (adjusted to pH 7.3, and 300 mOsm). All recordings were sampled at 10 kHz.

### Pharmacology

For measurement of the direct optical response of ChR2$^+$ and Chrim$^+$ pyramidal neurons to their respective target wavelengths, glutamatergic synaptic blockers CNQX (HelloBio HB0205) and D-AP5 (HelloBio HB0225) were used at concentrations of 40 μM and 80 μM, respectively.

### Connectivity

Connectivity between Chrim$^+$ and Opsin$^-$ pyramidal neurons was assessed through simultaneous current-clamp recordings where alternating trains of current were applied to each cell to generate multiple action potentials. A connection was considered to exist if the average excitatory post-synaptic potential (EPSP) amplitude was at least 3 times the baseline standard deviation. The first EPSP amplitude was calculated as the peak voltage of the EPSP subtracted by the baseline.

### Dual-targeted recordings

For simultaneous current-clamp recordings of ChR2$^+$ and Chrim$^+$ pyramidal neurons, neurons were identified by either the individual presence of EYFP for ChR2$^+$ or tdTomato for Chrim$^+$ neurons, and

additionally confirmed by the presence of a direct light-evoked response. Opsin-negative pyramidal neurons were identified by morphology, electrophysiological properties, and the lack of a direct optical response.

## Testing of learned dynamics with optogenetics

Following training, whole-cell current-clamp recordings were obtained from both ChR2$^+$ and Opsin$^-$ pyramidal neurons in slices from Early- and Late-trained groups. Optical stimulation during testing was administered using a dual-channel RGB LED (Vollong part #: VL-H01RGB00302), which projected red (625 nm) or blue (455 nm) light through the base of the recording chamber covering approximately a 1 mm diameter at the location of the recorded neurons. Trains of red and blue light pulses delivered during testing were identical in both structure and intensity as light delivered during training in the incubator. During testing, red light trains (625 nm, 12 pulses, 5 ms duration, at 25 Hz, intensity 0.2 mW/mm²) were delivered every 20 s, preceded by a 1 s baseline recording period per sweep. Evoked neuronal responses were analyzed in the 1 s window following the onset of red light stimulation. Recorded sweeps exhibiting spontaneous network activity during the baseline period were systematically excluded due to contamination of the light-evoked responses from spontaneous network activity.

Evoked neuronal activity was analyzed using spike-filtered voltage data, smoothed with a 10 ms moving average to reduce noise and improve the detection of voltage peaks and slopes. For each neuron, the median peak time of evoked activity was determined across a minimum of ten evoked sweeps and was calculated as the time point at which the peak voltage occurred during each sweep. The median event times of evoked activity were defined by identifying when the slopes of the voltage exceeded a threshold set at three times the standard deviation of all the slopes (calculated with a sliding window of 10 ms) and then taking the median of all these events. The center of gravity for each evoked trace was computed based on the midpoint of the integrated trace area. Cumulative distribution functions of evoked peak times were generated using the first 5 evoked sweeps from each neuron within both Early- and Late-training conditions. Analyses were performed blind to the training condition to prevent bias in the evaluation of evoked activity.

## Spontaneous event quantification/analysis

When spontaneous data was collected, a minimum of 5 min of activity was recorded for each neuron. Spontaneous network events were quantified based on previously defined criteria[17,52,73]. Spontaneous events were detected with a 5 mV voltage threshold above the resting membrane potential. However, during network events, the membrane potential would often make multiple crossings above and below the 5 mV threshold before returning to resting potential. Thus, we defined spontaneous events as activity that remained above the threshold for at least 100 ms, allowing for drops below the threshold that lasted less than 25 ms. The baseline was defined as a 100 ms period preceding the onset of a spontaneous event, and the analysis window was defined as 1 second following the detected onset of a spontaneous event. Detected spontaneous events were down-sampled by a factor of 10 to filter out spikes while preserving the overall temporal dynamics. Spontaneous median peak time, median event time, and mean center of gravity were computed in the same fashion as the evoked activity analysis. Cumulative distribution functions of evoked and spontaneous peak times were generated using the first 5 evoked sweeps or the first 5 spontaneous events from each neuron within both Early- and Late-training conditions.

## Correlation analysis of evoked and spontaneous activity

To assess the correlation between mean evoked and mean spontaneous activity within and between Early- and Late-training conditions,

evoked and spontaneous activity data were detected based on the same criterium as above. For each neuron, evoked activity data was temporally shifted by adjusting for the lag time observed before a 5 mV threshold crossing during evocation, ensuring the alignment of evoked neuronal responses across sweeps (this is the same threshold criterium used spontaneous event detection). Importantly, this time shift of the evoked activity was only used for the correlation between evoked and spontaneous activity and was not used to align evoked events for any other analyses used in this study.

Correlations between mean evoked and mean spontaneous activity was calculated to determine the degree of similarity in neural activity patterns within (intra-condition) and between (inter-condition) training conditions (Early and Late). The correlation matrices were populated by computing the Pearson correlation coefficient between mean traces of each combination of evoked and spontaneous activity. Each matrix element thus represented the correlation between a specific pair of aligned mean evoked and mean spontaneous activity traces. Diagonal elements of the intra-condition matrices, which represented self-correlations (mean evoked vs mean spontaneous within the same cell), were excluded to focus analysis on inter-slice correlations. The mean correlations were calculated for each comparison, emphasizing the generalized response pattern within each training condition compared to between training conditions.

## Pairwise correlation and cross-correlation analysis

Spontaneous events for pairwise correlations of activity dynamics between simultaneously recorded ChR2$^+$ and Chrim$^+$ pyramidal neurons were detected using the same method described earlier in the Methods. Spontaneous events were median-filtered and down-sampled by a factor of 50. Since detected event durations were not exactly the same between two simultaneously recorded neurons, we used a segment of the combined detected event index to compute the pairwise correlations of spontaneous event dynamics. Each segment spanned 1100 ms, including 100 ms of baseline activity before the start of the detected event.

The cross-correlations of spontaneous event bouts were computed using the same spike-filtered data as the pairwise correlations. However, to exclude transitions between active and inactive states, the analyzed voltage data were specifically sampled from 50 ms after the detected onset to 50 ms before the detected offset of each event.

## Prediction error analysis

To identify neurons exhibiting timed prediction error responses, we quantified the mean difference between responses to the conditioned stimulus (CS, red light alone) and conditioned + unconditioned stimulus (CSUS, red + blue light). For each neuron, the mean response to CS and CSUS was extracted, and the difference between these mean responses (CS - CSUS) was computed. This subtraction represents the neuron's differential response when exposed to CS alone versus CS paired with US. Neurons were classified as exhibiting a timed prediction error response based on the area under the curve of the mean CS - CSUS subtraction. Specifically, neurons for which this measure exceeded three standard errors of the mean above the average were identified as exhibiting timed prediction error, indicating a significant difference in their response to CS alone versus CS paired with US.

## Statistics and reproducibility

No statistical method was used to predetermine sample size for this study. The only data excluded from the analyses were from poor quality, unhealthy, or unstable whole-cell patch clamp recordings. Organotypic slices were prepared from mice without discrimination of sex, and slices were handled equivalently. Control slices were obtained from the same mice as the slices that underwent training and were balanced without bias. Seven different batches of organotypic slices were prepared from a total of 25–30 mice over the course of a year and trained under

equivalent conditions. Replications were successful, and recordings from these separate batches yielded consistent results. Investigators were not blind to the training condition during data collection, since the investigator would start and stop the training for each organotypic slice before recording on the rig. However, investigators were blind to the training conditions during the analysis of the data.

## Reporting summary

Further information on research design is available in the Nature Portfolio Reporting Summary linked to this article.

## Data availability

The raw data generated from patch-clamp recordings, which represent the central findings in this study, have been deposited in The Open Science Framework database along with the code for generating Fig. 2: https://osf.io/qspnv/. Source data for Figs. 2–6 and supplemental figures are provided with this paper in the Source Data file. Source data are provided in this paper.

## Code availability

Code used for data analysis and visualization of results have been deposited in The Open Science Framework database: https://osf.io/qspnv/.

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

## Acknowledgements

We would like to thank Saray Soldado Magraner, Jeffrey Yang, Kloey Reyes, and Kayley Steele for helpful scientific discussions and technical assistance. This work was supported by the National Institute of Neurological Disorders and Stroke grant NS116589 and CRCNS grant NS125877.

## Author contributions

B.L. and D.V.B. designed research; B.L. performed research and analyzed the data; B.L. wrote the first draft of the paper; B.L. and D.V.B. edited the paper; B.L. and D.V.B. wrote the paper.

## Competing interests

The authors declare no competing interests.
