## [Transparent Peer Review file · Nature Communications]

Ex Vivo Cortical Circuits Learn to Predict and Spontaneously Replay Temporal Patterns

Corresponding Author: Dr Dean Buonomano

Version 0:

Reviewer comments:

Reviewer #1

(Remarks to the Author)

I enjoyed reading this compelling report of temporal prediction in ex vivo preparations and its interpretation in relation to predictive coding accounts of neuronal self-organisation. I thought your study and analyses were motivated clearly and that the results and conclusions were compelling. My only suggestion is to foreground some of your key findings and link them to relevant issues in the systems and cognitive neuroscience literature. Perhaps you could consider the following:

Major points

Your analysis of prediction error-like responses (i.e., the comparison of red alone and red – blue responses) could be foregrounded more in the main text and abstract. This is a potentially remarkable finding that would engage many people in predictive processing and cognitive neuroscience. I would recommend something like the following in the main text (and a nod to it in the abstract):

"The identification of single neurons that show prediction error responses is among the first lines of definitive empirical evidence for predictive coding in the brain. Hitherto, predictive coding has relied upon non-invasive electrophysiology: for example, using the mismatch negativity paradigm [1, 2]. On this view, one can interpret the above responses as omission-related responses. In other words, neuronal responses to the absence of a predicted (blue) stimulus. Crucially, these responses were evinced in well-characterised single (excitatory, pyramidal) neurons."

A similar emphasis could apply to the asymmetry of the connections. This asymmetry is garnering more attention in systems neuroscience and your results could be usefully connected to this literature. For example:

"The asymmetry in connectivity between excitatory subpopulations speaks to a general theme in the self-organisation of neuronal (and biotic) dynamics; namely, the breaking of detailed balance and the emergence of non-dissipative or solenoidal dynamics. In short, asymmetry in the coupling of sparsely connected systems is necessary for the itinerant behaviour that characterises biotic self-organisation; ranging from oscillations through to lifecycles. This is evident at nearly every scale through from turbulence in cortical field characterisations through to Red-Queen dynamics in evolution [3-7]. Indeed, much of the empirical evidence for predictive coding rests upon functional asymmetries in the exchange between prediction and prediction error units [8-11]."

Finally, I think you can generalise the discussion of your conclusions beyond the (colloquial) framing of predictive coding. While it is true that most hierarchical predictive coding schemes emphasise the role of bottom-up and top-down influences, general formulations include lateral interactions [12, 13]; especially when considering the role of neuromodulation and precision (i.e., excitation and inhibition balance). For example, see [8, 14]. If you generalise predictive coding to include predictive processing (a.k.a., active inference) there are accounts that would accommodate your findings in a non-hierarchical setting. See for example [15]. This author has also done some interesting in vitro work interpreting neuronal responses in terms of active inference and free energy (i.e., prediction error) minimisation.

Minor points

In the section on asymmetric connectivity, you note that there was a difference in connection strength but no difference in connection probability. It would be useful to have operational definitions of these connection metrics in the main text. For example, was connection strength defined in terms of cross correlations between two neurons? Does connection probability refer to the probability of there being a significant functional connectivity in a randomly selected pair of neurons?

I hope that these comments help should any revision be required.

1. Walsh, K.S., et al., Evaluating the neurophysiological evidence for predictive processing as a model of perception. *Ann N Y Acad Sci*, 2020. 1464(1): p. 242-268.
2. Hodson, R., M. Mehta, and R. Smith, The empirical status of predictive coding and active inference. *Neurosci Biobehav Rev*, 2024. 157: p. 105473.
3. Deco, G. and M.L. Kringelbach, Turbulent-like Dynamics in the Human Brain. *Cell Rep*, 2020. 33(10): p. 108471.
4. Gu, S., et al., The Energy Landscape of Neurophysiological Activity Implicit in Brain Network Structure. *Sci Rep*, 2018. 8(1): p. 2507.
5. Jiang, D.-Q., M. Qian, and M.-P. Qian, Mathematical theory of nonequilibrium steady-states. *Lecture Notes in Mathematics*. 2004: Springer.
6. Zhang, F., et al., The potential and flux landscape theory of evolution. *J Chem Phys*, 2012. 137(6): p. 065102.
7. Ao, P., Global view of bionetwork dynamics: adaptive landscape. *Journal of Genetics and Genomics*, 2009. 36(2): p. 63-73.
8. Bastos, A.M., et al., Canonical microcircuits for predictive coding. *Neuron*, 2012. 76(4): p. 695-711.
9. Bastos, A.M., et al., A DCM study of spectral asymmetries in feedforward and feedback connections between visual areas V1 and V4 in the monkey. *Neuroimage*, 2015. 108: p. 460-75.
10. Arnal, L.H. and A.L. Giraud, Cortical oscillations and sensory predictions. *Trends Cogn Sci*, 2012. 16(7): p. 390-8.
11. Hovsepian, S., I. Olasagasti, and A.-L. Giraud, Combining predictive coding with neural oscillations optimizes on-line speech processing. *bioRxiv*, 2018: p. 477588.
12. Bogacz, R., A tutorial on the free-energy framework for modelling perception and learning. *J Math Psychol*, 2017. 76(Pt B): p. 198-211.
13. Salvatori, T., et al. Reverse Differentiation via Predictive Coding. 2021. *arXiv:2103.04689*.
14. Shipp, S., Neural Elements for Predictive Coding. *Front Psychol*, 2016. 7: p. 1792.
15. Isomura, T., Active inference leads to Bayesian neurophysiology. *Neuroscience Research*, 2022. 175: p. 38-45.

Reviewer #2

(Remarks to the Author)

Review.

This manuscript studies an intrinsic property of the cortical microcircuits to encode temporal information in addition to the predictions and prediction errors. The provided results demonstrate strong evidence that cortical microcircuits have the inherent capacity to learn and predict temporal patterns of external stimulation. By using a combination of the cortical organotypic slice preparation with dual-optical stimulation, the authors demonstrate that after 24 hours of training, the cortical networks change their dynamics to match the specific temporal patterns. The observation of replay during spontaneous activity and the demonstration of temporal predictions and prediction errors support the statement that these microcircuits can autonomously generate internal predictions.

This work is important because it advances our understanding of the computations performed by the local cortical microcircuits. The results show that temporal learning and spontaneous replay can arise independently of the top-down projections are novel and support the previously published computational models of predictive coding.

There are several strengths: innovative use of cortical organotypic slices combined with dual-optical stimulation; the study is designed to test the computational predictive coding hypothesis. Another important mechanistic insight into how the learning occurs is the newly described asymmetric connectivity between distinct neuronal ensembles. It would be interesting to see how it could be related to potential asymmetric connectivity within the brain. Description of spontaneous replay is also interesting.

Although the manuscript is strong overall, there are several questions left:

- 1) The work is done in slice cultures, which is known to form a lot of recurrent connections that may lead to the emergence of spontaneous bursts of activity. It remains unclear how these results translate to *in vivo* conditions with more complex dynamics. The manuscript could benefit from a discussion on the potential limitations of generalizing these findings to the intact brain.
- 2) The description of timed predictions and prediction errors is very interesting, but the manuscript provides limited analysis of this phenomenon. A more detailed information of where the prediction and prediction error cells are located in slice and what is their type would be valuable.
- 3) One of the main computations of the predictive coding theory is subtraction of the sensory stimuli from the predictions generating prediction errors. This computation is hypothesized to be performed by the inhibitory neurons. Out of the patched cells it is likely that there were some inhibitory neurons. It would be interesting to see how their activity looks like and what is their relationship to the prediction and prediction error cells.

We thank the reviewers for their assessments that *“I enjoyed reading this compelling report of temporal prediction in ex vivo preparations and its interpretation in relation to predictive coding accounts of neuronal self-organisation”* and *“The provided results demonstrate strong evidence that cortical microcircuits have the inherent capacity to learn and predict temporal patterns of external stimulation”*. We also thank them for their very constructive comments and suggestions. In response to these suggestions, we have modified the text and performed additional analyses.

Below we provide a detailed response to the reviewers’ comments and an explanation of all the changes made to the manuscript (all reviewer comments are in italics, and our responses are in bold).

REVIEWER 1

I enjoyed reading this compelling report of temporal prediction in ex vivo preparations and its interpretation in relation to predictive coding accounts of neuronal self-organisation. I thought your study and analyses were motivated clearly and that the results and conclusions were compelling. My only suggestion is to foreground some of your key findings and link them to relevant issues in the systems and cognitive neuroscience literature. Perhaps you could consider the following:

We thank the reviewer for the positive assessment and the insightful suggestions.

R1.1. *Your analysis of prediction error-like responses (i.e., the comparison of red alone and red – blue responses) could be foregrounded more in the main text and abstract. This is a potentially remarkable finding that would engage many people in predictive processing and cognitive neuroscience. I would recommend something like the following in the main text (and a nod to it in the abstract):*

“The identification of single neurons that show prediction error responses is among the first lines of definitive empirical evidence for predictive coding in the brain. Hitherto, predictive coding has relied upon non-invasive electrophysiology: for example, using the mismatch negativity paradigm [1, 2]. On this view, one can interpret the above responses as omission-related responses. In other words, neuronal responses to the absence of a predicted (blue) stimulus. Crucially, these responses were evinced in well-characterised single (excitatory, pyramidal) neurons.”

We have now incorporated additional text in the Abstract and Discussion highlighting the points above. We used similar wording, but wanted to be careful to not potentially overstate our finding regarding “definite empirical evidence”. The comparison to omission responses is an excellent point which we now mention as well.

Page 2, Line 10-11: “Furthermore, some neurons exhibited timed prediction errors as revealed by larger responses when the expected stimulus was omitted compared to when it was present.”

Page 12, Line 31-36: “The identification of single neurons that show prediction error responses provides direct evidence for predictive coding in a reduced and highly controlled cortical network. The large response observed to the omission of the expected blue light can also be characterized

as an omission response. And the fact that these responses are delayed, peaking approximately 265 ± 39 ms after the expected onset of the blue light, further indicated that these responses rely on complex changes in the internal circuitry rather than simple blue-light inhibition.”

R1.2. *A similar emphasis could apply to the asymmetry of the connections. This asymmetry is garnering more attention in systems neuroscience and your results could be usefully connected to this literature. For example:*

“The asymmetry in connectivity between excitatory subpopulations speaks to a general theme in the self-organisation of neuronal (and biotic) dynamics; namely, the breaking of detailed balance and the emergence of non-dissipative or solenoidal dynamics. In short, asymmetry in the coupling of sparsely connected systems is necessary for the itinerant behaviour that characterises biotic self-organisation; ranging from oscillations through to lifecycles. This is evident at nearly every scale through from turbulence in cortical field characterisations through to Red-Queen dynamics in evolution [3-7]. Indeed, much of the empirical evidence for predictive coding rests upon functional asymmetries in the exchange between prediction and prediction error units [8-11].”

Thank you for stressing the importance of the asymmetric connectivity, which indeed has broad implications across complex systems. We now also highlight this point, but we felt that explicitly mentioning solenoidal dynamics, Red-Queen dynamics, or turbulence, might require an explanation of these phenomenon as the typical neuroscience reader might not be familiar with them. But we do cite a number of the suggested papers.

R1.3. *Finally, I think you can generalise the discussion of your conclusions beyond the (colloquial) framing of predictive coding. While it is true that most hierarchical predictive coding schemes emphasise the role of bottom-up and top-down influences, general formulations include lateral interactions [12, 13]; especially when considering the role of neuromodulation and precision (i.e., excitation and inhibition balance). For example, see [8, 14]. If you generalise predictive coding to include predictive processing (a.k.a., active inference) there are accounts that would accommodate your findings in a non-hierarchical setting. See for example [15]. This author has also done some interesting in vitro work interpreting neuronal responses in terms of active inference and free energy (i.e., prediction error) minimisation.*

In the Discussion we have now also emphasized the important issue of top-down prediction error signals in comparison with our current results that support that local microcircuits seem to be autonomously able to generate prediction errors in the absence of top-down circuits.

Page 13, Line 22-27: “As mentioned above, most theories of predictive coding propose that prediction error signals rely on feedback from higher-order cortical areas (Bastos et al., 2012; Shipp, 2016; Bogacz, 2017; Rao, 2024). The current results support the possibility that each local neocortical microcircuit is autonomously able to generate prediction errors (Isomura, 2022)—which, of course, does not imply that feedback is not also critical. And, of course, it remains to be determined if local neocortical circuits *in vivo* are autonomously capable of learning temporal predictions in the absence of interareal and top-down connections.”

Minor points

In the section on asymmetric connectivity, you note that there was a difference in connection strength but no difference in connection probability. It would be useful to have operational definitions of these connection metrics in the main text. For example, was connection strength defined in terms of cross correlations between two neurons? Does connection probability refer to the probability of there being a significant functional connectivity in a randomly selected pair of neurons?

We now clarify that this statement is based simply on the mean synaptic strength of the monosynaptically connected pairs. Given the limited number of recordings, we are constrained regarding general statements about synaptic strength distribution, and whether we were perhaps underpowered to detect changes in connectivity.

Page 13, Line 16-19: “Based on the average synaptic strength of the monosynaptically connected pairs, we observed an asymmetry between the Chrim⁺ and Opsin⁻ pyramidal subpopulations: synaptic connections were stronger in the Chrim⁺→Opsin⁻ direction compared to the Opsin⁻→Chrim⁺ direction (Fig. 6).”

REVIEWER 2

This manuscript studies an intrinsic property of the cortical microcircuits to encode temporal information in addition to the predictions and prediction errors. The provided results demonstrate strong evidence that cortical microcircuits have the inherent capacity to learn and predict temporal patterns of external stimulation. By using a combination of the cortical organotypic slice preparation with dual-optical stimulation, the authors demonstrate that after 24 hours of training, the cortical networks change their dynamics to match the specific temporal patterns. The observation of replay during spontaneous activity and the demonstration of temporal predictions and prediction errors support the statement that these microcircuits can autonomously generate internal predictions. This work is important because it advances our understanding of the computations performed by the local cortical microcircuits. The results show that temporal learning and spontaneous replay can arise independently of the top-down projections are novel and support the previously published computational models of predictive coding. There are several strengths: innovative use of cortical organotypic slices combined with dual-optical stimulation; the study is designed to test the computational predictive coding hypothesis. Another important mechanistic insight into how the learning occurs is the newly described asymmetric connectivity between distinct neuronal ensembles. It would be interesting to see how it could be related to potential asymmetric connectivity within the brain. Description of spontaneous replay is also interesting.

Although the manuscript is strong overall, there are several questions left:

We thank the reviewer for discerning comments and thoughtful summary of the paper.

R2.1 *The work is done in slice cultures, which is known to form a lot of recurrent connections that may lead to the emergence of spontaneous bursts of activity. it remains unclear how these results translate to in vivo conditions with more complex dynamics. The manuscript could*

benefit from a discussion on the potential limitations of generalizing these findings to the intact brain.

It is indeed the case that *de novo* synaptogenesis occurs in organotypic slices, which is believed to reflect the ability of these networks to return to their homeostatic setpoints—as expected of any self-organizing/self-correcting system. To the best of our knowledge there is no evidence that any synaptogenesis is “inappropriate” (Bolz et al., 1990; Gähwiler et al., 1997; Dantzker and Callaway, 1998; De Simoni et al., 2003; Hanson and Madison, 2007). For example, autapses (as can be observed in dissociated cultures) or connections between incorrect cell types have never been reported—indeed organotypic slices have recently been used for circuit breaking in human neocortex and the connectivity observed in the organotypic and acute slices was similar (Kim et al., 2023). Additionally, as the reviewer pointed out, organotypic slices generate spontaneous bouts of activity. We have carefully studied these network events and there is significant agreement that they correspond to the Up-states observed *in vivo* (Johnson and Buonomano, 2007; Goel and Buonomano, 2013; Motanis and Buonomano, 2015; Romero-Sosa et al., 2021).

We, of course, agree with the reviewer, that it remains to be seen if *in vivo* local neocortical circuits also autonomously generate prediction errors. We now point this out in the Discussion—however, this is a very challenging question to answer *in vivo*, which is one of the advantages of organotypic cultures. But we believe we have taken an important first step towards establishing that at least under some conditions neocortical circuits are autonomously capable of learning temporal predictions.

Page 13, Line 22-27: “As mentioned above, most theories of predictive coding propose that prediction error signals rely on feedback from higher-order cortical areas (Bastos et al., 2012; Shipp, 2016; Bogacz, 2017; Rao, 2024). The current results support the possibility that each local neocortical microcircuit is autonomously able to generate prediction errors (Isomura, 2022)—which, of course, does not imply that feedback is not also critical. And, of course, it remains to be determined if local neocortical circuits *in vivo* are autonomously capable of learning temporal predictions in the absence of interareal and top-down connections.”

We also further address the organotypic preparation in the Introduction.

Page 3, Line 26-41: “Indeed, spontaneous activity in neocortical organotypic slices converges to similar dynamic regimes observed *in vivo*. Specifically, they exhibit Up-state/Down-state transitions that reflect the well-balanced excitatory-inhibitory regimes critical to normal neocortical function (Johnson and Buonomano, 2007; Romero-Sosa et al., 2021; Sadeh and Clopath, 2021; Bak et al., 2024). It is well established that synaptogenesis occurs in organotypic slices, which is thought to be driven by homeostatic learning rules aimed at bringing network dynamics back to homeostatic setpoints (Bolz, 1994; Echevarria and Albus, 2000; Humpel, 2015; Bak et al., 2024). There is no evidence of abnormal connectivity between cell types or of autapses (as observed in dissociated cultures)—indeed, recent studies of microcircuit connectivity in human neocortex have revealed similar connection patterns in organotypic and acute cultures of human neocortex. Additionally, the synaptic learning rules observed in acute slices and *in vivo* are present in organotypic slices, indeed a number of early studies of synaptic plasticity were performed in organotypic cultures (Debanne et al., 1994; Musleh et al., 1997; Hayashi et al., 2000; Barria and Malinow, 2002; Goold and Nicoll, 2010; Yamada et al., 2010; Letellier et al., 2019; Anisimova et al., 2022). Finally, and critical to our goals here, organotypic slices coupled with optogenetics provide a tractable way to fully control “sensory” experience and to study forms of learning that

may take hours or days to develop.”

R2.2 *The description of timed predictions and prediction errors is very interesting, but the manuscript provides limited analysis of this phenomenon. A more detailed information of where the prediction and prediction error cells are located in slice and what is their type would be valuable.*

Indeed, we are very excited to pursue the question of potential differential effects/computational roles of different neuronal subtypes. In the current work all recordings were performed in Layer II/III from labeled and putative pyramidal neurons. Thus we are somewhat limited in our ability to address this issue with the current data. Furthermore, because our approach uses both a blue and red opsins with their respective reporters, we are somewhat constrained in terms of using standard molecular/genetic markers of specific subpopulations of neurons.

But since numerous subtypes of pyramidal neurons also differ in their intrinsic properties, we have reanalyzed our recordings and examined whether there was any correlation between intrinsic neuronal properties (input resistance, membrane time constants, and excitability) and if the neurons exhibited prediction errors or not. As shown in Figure R1, we did not observe significant differences in the membrane time constants or input resistances of the neurons that exhibited timed prediction errors and those that did not. There was a small non-significant difference in the intrinsic excitability at the highest intensities. Even if this effect was significant, subsequent experiments would require disentangling whether any such effect reflected different subtypes of pyramidal neurons or experience-dependent changes homeostatically driven by their differential firing rates.

Page 11, Line 21-25: “There was no significant difference in the input resistance (182 ± 11 and 187 ± 14 M Ω), membrane time constant (10.4 ± 0.67 and 10.3 ± 0.37 ms), or intrinsic excitability (number of spikes per current step ranging from 0.05-0.3 nA) between the neurons that exhibited timed prediction errors and those that did not. But future studies will have to examine whether the observed distinction between prediction error and non-prediction error neurons correspond to specific excitatory neuron subtypes.”

Figure R1. A. Input resistance between subpopulations of Opsin⁻ pyramidal neurons that exhibited timed prediction error and those that did not (see **Figure 7** of paper) following Late-training (182 ± 11 and $187 \pm 14 M\Omega$, respectively; $p = 0.85$; Timed Error $n = 5$; No Error $n = 17$). **B.** Membrane time constant between subpopulations of Opsin⁻ pyramidal neurons that exhibited timed prediction error and those that did not following Late-training (10.4 ± 0.67 and 10.3 ± 0.37 ms, respectively; $p = 0.92$; Timed Error $n = 5$; No Error $n = 17$). **C.** F-I (input-output) curves of the Opsin⁻ pyramidal neurons that exhibited a temporal prediction error and those that did not following Late-training (current steps 0.05-0.3 nA; $p > 0.05$; Timed Error $n = 5$; No Error $n = 17$). Values above each pair of data points correspond to uncorrected p-values of t-tests.

R2.3 One of the main computations of the predictive coding theory is subtraction of the sensory stimuli from the predictions generating prediction errors. This computation is hypothesized to be performed by the inhibitory neurons. Out of the patched cells it is likely that there were some inhibitory neurons. It would be interesting to see how their activity looks like and what is their relationship to the prediction and prediction error cells.

Yes, virtually all predictive coding models predict an important role for inhibitory neurons. It will be critical to determine if the learning itself relies on inhibitory plasticity, and if PV, SST, or other inhibitory subtypes are primarily responsible for the inhibition during successfully predicted stimuli. During our recordings we actively avoided apparent inhibitory neurons. Re-examination of our data set did not reveal any obvious conventional fast-spiking PV neurons that are fairly easy to distinguish based on electrophysiological properties (Romero-Sosa et al., 2021). However, it is possible that some of the neurons we recorded were SST inhibitory neurons—which are harder to identify based on the F-I curves. Unfortunately, as mentioned above, a current shortcoming of our dual-opsin approach is that it limits our ability to use the red channel for the standard identification of inhibitory neuron subtypes.

- Bolz J, Novak N, Gotz F, Bonhoeffer T (1990) Formation of target-specific neuronal projections in organotypic slices cultures from rat visual cortex. *Nature* 346:359-562.
- Dantzker JL, Callaway EM (1998) The development of local, layer-specific visual cortical axons in the absence of extrinsic influences and intrinsic activity. *J Neurosci* 18:4145-4154.
- De Simoni A, Griesinger CB, Edwards FA (2003) Development of rat CA1 neurones in acute versus organotypic slices: role of experience in synaptic morphology and activity. *J Physiol* 550:135-147.
- Gähwiler BH, Capogna M, Debanne D, McKinney RA, Thompson SM (1997) Organotypic slice cultures: a technique has come of age. *Trends Neurosci* 20:471-477.
- Goel A, Buonomano DV (2013) Chronic electrical stimulation homeostatically decreases spontaneous activity, but paradoxically increases evoked network activity. *Journal of Neurophysiology* 109:1824-1836.
- Hanson JE, Madison DV (2007) Presynaptic FMR1 genotype influences the degree of synaptic connectivity in a mosaic mouse model of fragile X syndrome. *J Neurosci* 27:4014-4018.
- Johnson HA, Buonomano DV (2007) Development and Plasticity of Spontaneous Activity and Up States in Cortical Organotypic Slices. *J Neurosci* 27:5915-5925.
- Kim M-H et al. (2023) Target cell-specific synaptic dynamics of excitatory to inhibitory neuron connections in supragranular layers of human neocortex. *eLife* 12:e81863.
- Motanis H, Buonomano DV (2015) Delayed in vitro Development of Up States but Normal Network Plasticity in Fragile X Circuits. *Eur J Neurosci* 42:2312-2321.
- Romero-Sosa JL, Motanis H, Buonomano DV (2021) Differential Excitability of PV and SST Neurons Results in Distinct Functional Roles in Inhibition Stabilization of Up States. *The Journal of Neuroscience* 41:7182-7196.